# Implementation of maternal and perinatal death reviews: a scoping review protocol

Mary V Kinney ,[1] David Roger Walugembe,[2] Phillip Wanduru,[3] Peter Waiswa,[3,4] Asha S George[1]

[1]School of Public Health, Faculty of Community and Health Sciences, University of the Western Cape, Cape Town, Bellville, South Africa
[2]School of Health Studies, Faculty of Information and Media Studies, University of Western Ontario, London, Ontario, Canada
[3]School of Public Health, Makerere University College of Health Sciences, Kampala, Uganda
[4]Division of Global Health, Department of Public Health Sciences, Karolinska Institutet, Stockholm, Sweden

**Correspondence to**
Mary V Kinney;
mkinney@uwc.ac.za

## ABSTRACT

**Introduction** Maternal and perinatal death surveillance and response (MPDSR), or any related form of audit, is a systematic process used to prevent future maternal and perinatal deaths. While the existence of MPDSR policies is routinely measured, measurement and understanding of policy implementation has lagged behind. In this paper, we present a theory-based conceptual framework for understanding MPDSR implementation as well as a scoping review protocol to understand factors influencing MPDSR implementation in low/ middle-income countries (LMIC).

**Methods and analysis** The Consolidated Framework for Implementation Research will inform the development of a theory-based conceptual framework for MPDSR implementation. The methodology for the scoping review will be guided by an adapted Arksey and O'Malley approach. Documents will include published and grey literature sourced from electronic databases (PubMed, CINAHL, SCOPUS, Web of Science, JSTOR, LILACS), the WHO Library, Maternal Death Surveillance and Response Action Network, Google, the reference lists of key studies and key experts. Two reviewers will independently screen titles, abstracts and full studies for inclusion. All discrepancies will be resolved by an independent third party. We will include studies published in English from 2004 to July 2018 that present results on factors influencing implementation of MPDSR, or any related form. Qualitative content and thematic analysis will be applied to extracted data according to the theory-based conceptual framework. Stakeholders will be consulted at various stages of the process.

**Ethics and dissemination** The scoping review will synthesise implementation factors relating to MPDSR in LMIC as described in the literature. This review will contribute to the work of the Countdown to 2030 Drivers Group, which seeks to explore key contextual drivers for equitable and effective coverage of maternal and child health interventions. Ethics approval is not required. The results will be disseminated through various channels, including a peer-reviewed publication.

## INTRODUCTION

Most deliveries in developing countries now happen in hospitals and clinics making facility-based maternal and newborn care a global health imperative for achieving

## Strengths and limitations of this study

► To our best knowledge, this protocol describes the first scoping review to identify and describe implementation factors relating to maternal and perinatal death surveillance and response (MPDSR) in low-income and middle-income countries.
► The search strategy includes six electronic databases with peer-reviewed literature as well as three online search engines to identify published and grey literature including academic research articles, commentaries, other related reviews and reports.
► Qualitative thematic and content analysis will be used to analyse the data linked to an adapted theory-based conceptual framework for MPDSR implementation.
► Key stakeholders will be consulted and engaged throughout the study review process, including the World Health Organization's MPDSR Technical Working Group as well as the Countdown to 2030 Drivers Technical Working Group.
► Limitations relate to the search criteria, notably around language (English only) and time span (from 2004-July 2018) as well as the search process (eg, not all grey literature can be identified).

the sustainable development goal for health.[1 2] Maternal death surveillance and response (MDSR), perinatal death audit or a joint maternal and perinatal death surveillance and response (MPDSR) is one process used to prevent maternal and perinatal deaths.[3–5] Maternal and/or perinatal death surveillance and response (M/PDSR) is an established mechanism to examine the circumstances surrounding each death to prevent future deaths.[6] It requires continuous application of monitoring–review–act cycles[7] to capture information on the number and causes of deaths, with systematic, critical analysis of the care received for a sample of or for all cases, in a no-blame, interdisciplinary setting, with a view to improving the care provided to all mothers and babies.[8] The potential for MPDSR to improve mortality

outcomes only occurs if the audit cycle is completed and implemented overtime triggering iterative cycles of improvement.[9–11]

In the past 15 years, there has been momentum to strengthen clinical audit practice for maternal and perinatal deaths,[12–15] including the development of global technical guidelines.[8 16] Many low/middle-income countries (LMIC) have adopted national guidelines, however, few have robust MPDSR systems.[9] A growing number of studies have investigated the implementation of M/PDSR in selected countries, and some reviews have explored implementation factors for maternal death reviews or perinatal death audits, separately. For example, a structured literature review of accountability mechanisms for maternal and newborn health in sub-Saharan Africa found MDSR the most common mechanism for performance accountability.[17] A systematic review of facility-based perinatal mortality audit in LMIC in 2009 identified 10 low-quality evaluations with mortality outcome data.[10] A literature review conducted in 2015 on facility-based perinatal audits explored enablers and barriers according to the health system building blocks.[9]

While there are valuable contributions to the literature, these previous reviews did not consider implementation theory to assess implementation of M/PDSR nor of the full range of types of maternal and/or perinatal death reviews.[9 17 18] Implementation theory allows for more complex interventions to be unpacked and examined.[19–24] This approach enables exploration of issues, such as trust, credibility, relationships and hierarchies to understand factors that support or hinder implementation.[25] Interventions seeking to improve facility-based care are often ongoing processes that are complex, fluid and context specific.[7 19 24 26] A variety of factors,[19–24 27] including context,[28] can influence implementation of these types of interventions. With rising attention on facility-based maternal and newborn healthcare,[5 29–31] more needs to be understood about the implementation of M/PDSR.

### Study rationale

Global agencies, such as the WHO, have created guidelines for M/PDSR and are encouraging LMIC to move forward with implementation.[8 14] Further understanding of the enablers and barriers of implementation in LMIC is needed to support roll out of this intervention across and within countries. A rigorous, scoping review has not yet been undertaken to map publications in LMIC on factors influencing implementation of M/PDSR in ways that are inclusive of either maternal and/or perinatal death audits.

### Study objectives

To map and synthesise the available literature on the factors that support or hinder M/PDSR implementation using a theory-based conceptual implementation framework. We will also explore common, if any, implementation factors among MDSR, PDSR or MPDSR.

## METHODS AND ANALYSIS
### Conceptual model

This section of the protocol presents a proposed theory-based conceptual framework, which will be pilot tested and adapted for the data extraction and analysis. To develop the framework, we considered conceptualisation of the M/PDSR as an intervention process and reviewed various theory-based implementation frameworks.

### Conceptualising M/PDSR

M/PDSR is a continuous action cycle for quality improvement that links maternal and perinatal mortality data from the local to the national level. M/PDSR can be considered as an intervention as well as an implementation process since it is a set of efforts geared towards facilitating change.[25] At all levels, the process relies on the effective reporting and assigning causes to deaths, on identifying actions that may contribute to the prevention of further deaths, assigning those actions to particular groups or individuals within a specified timeframe and following up to ensure that those actions have been implemented. At the facility level, a six-step cycle of auditing deaths is recommended whereby: (1) cases for review are identified; (2) information on these cases is collected; (3) the information is analysed and discussed by the MPDSR committee; (4) solutions are recommended based on the findings of the analysis; (5) solutions are implemented and (6) feedback or reflection on if solutions were implemented and what worked or did not in order to inform the process moving forward.[8]

In a well-functioning health system, the information from the facility-level audits feeds up into a sub-national level process whereby information about maternal and perinatal deaths is received, compiled, reviewed for completeness and any relevant actions at that level or above. The information is further analysed and then disseminated to appropriate stakeholders, including other sub-national entities who would have their own processes (eg, district to province). Information from the sub-national level is compiled and sent to national level whereby further synthesis and analyses are conducted. This often leads to a national annual report that is then disseminated back to sub-national and facility levels.

As a concept, M/PDSR functions at multiple levels of the health system—national, sub-national and facility (and for some countries community level components are included in the process). The communication system and inter-connectedness between the different levels are an important component of M/PDSR since the process is a reporting mechanism moving continuously from bottom up—facility to national—and also from top down—national to facility. For example, recommendations to the national Ministry of Health could be identified during a facility-level audit process. This information should be fed up through the system to reach the national level decision makers. Likewise, the national level needs information from the facility level and sub-national level in order to assess the situation of maternal and perinatal

## Box 1    Conceptual implementation framework for M/PDSR

### Domain 1: Intervention/MPDSR
SERVICE DELIVERY LENS (tangible inputs)
Executing audit: steps of cycle implemented
Cost and funding for the audit process including collecting data, meeting related costs including transport, specific training, running secretariat, time
SOCIETAL LENS (social understanding and relationships)
Intervention source: legitimacy depending on whether intervention is externally or internally developed
Evidence strength and quality: evidence supporting the belief that the intervention will have desired outcomes (reduced mortality; changes undertaken to improve quality of care/'response')
Relative advantage: perception of the advantage of implementing the intervention versus an alternative solution
SYSTEMS LENS (change dynamics)
Trialability: ability to test/pilot the intervention on a small scale, learn and revise if warranted
Reflectivity: feedback about the progress and quality of implementation accompanied with regular personal and team debriefing about progress and experience
Adaptability: degree to which an intervention can be tailored to meet the needs of an organisation (core vs peripheral elements)
Complexity: perceived difficulty of implementation by the implementers (extent of disruption, number of elements/steps, extent of discretion, health system levels, actors)

### Domain 2: Outer setting/broader context
SERVICE DELIVERY LENS (tangible inputs)
Policy and planning: MPDSR policy and guidelines, death notification requirements (legal framework for notifying deaths), legal mandate, litigation/legal protection
Resource flows: any mention of funding support or resources for MPDSR (eg, sponsors, related costs being funded/budgeted)
SOCIETAL LENS (social understanding and relationships)
Linkages to other actors: local party, union affiliations, professional associations, community organisations
Pressure: to implement from actors and other implementers
Community links: awareness of MPDSR in the community (grassroots); community or CHW engagement and participation in MPDSR
SYSTEMS LENS (change dynamics)
Cosmopolitanism: level of connectedness and networks with other health system levels, organisations and therefore openness or resistance to change

### Domain 3: Inner setting
SERVICE DELIVERY LENS (tangible inputs)
Readiness to implement: committees formed, training, focal point identified, availability of tools, leadership engagement and management capacity, HRH workload, access to resources
Structural characteristics of social architecture (characteristics of the team, for example, size, interdisciplinary nature, membership regulation)
Incentives/rewards (disincentives/sanctions): extrinsic incentives such as goal-sharing awards, performance reviews/promotions, training, tea or the consequences
SOCIETAL LENS (social understanding and relationships)
Networks and communication: nature and quality of communication within audit team (including hierarchies, mentorship, teamwork)
Culture: norms and values, organisational assumptions (blame culture vs trust)
SYSTEMS LENS (change dynamics)

Continued

## Box 1    Continued

Implementation climate: explanation of environment, for example, learning climate, relative priority, if there are things mentioned that are tensions/triggers for change
Agents of change: individuals who have formal or informal influence on the attitudes and beliefs of their colleagues with respect to implementing the intervention or on the implementation process overall

### Domain 4: Individuals
SERVICE DELIVERY LENS (tangible inputs)
Technical skills and knowledge: individual staff knowledge and competencies
SOCIETAL LENS (social understanding and relationships)
Individual motivation, self-efficacy: an individual's confidence in their capabilities to execute the implementation; individuals who are motivated
Individual commitment/ownership to team and organisation: individuals' perception of their commitment to the organisation and their relationship
Individual commitment/ownership of intervention: individuals' perception of their commitment to the intervention
Individual orientation: personal traits such as tolerance of ambiguity, team player, flexibility, problem solving, critical thinking
SYSTEMS LENS (change dynamics)
Individual state of change: phase an individual is in as he or she progresses toward skilled, enthusiastic and sustained use of the intervention

mortality in the country in order to make recommendations at sub-national and facility levels.

### Theory-based implementation framework
We reviewed theories, models and frameworks consolidated by others as well as M/PDSR specific literature to determine a list of possible frameworks to consider.[6 27 32 33]

Lewis' commentary on MDSR argues the importance of considering different 'cultural factors' relating to M/PDSR including factors at the individual, institutional and policy levels.[6] For example, at a micro level, an individual's willingness to 'self-correct' requires commitment of staff towards conducting audit themselves, to accept open discussion with peers and to take forward the actions recommended. At a meso level, proactive institutional ethos that promotes learning as a critical part of quality improvement shapes the organisational culture. An environment open to learning also requires individual responsibility and ownership of the process, whereby clinicians need to improve their practice and change their behaviour for the betterment of maternal and perinatal health. A supportive policy and political environment (macro level) would need to be in place to initiate and support implementation.[6]

With the understanding that M/PDSR is an intervention process functioning at multiple levels of the health system, we identified five implementation frameworks for in-depth review and mapped their components with each other and in relation to M/PDSR (online supplementary file 1).[34–38] Our mapping process found that both the

Context and Implementation of Complex Intervention Framework[36] and the Dynamic Sustainability Framework[38] did not provide enough consideration to the implementation process of the intervention. The Promoting Action on Research Implementation in Health Services Framework[34] and the Normalisation Process Theory[39] had strengths especially at the meso and micro levels for understanding implementation processes; however, there was not enough overlap with the concepts identified for M/PDSR implementation. The Consolidated Framework for Implementation Research (CFIR) was found to be the most relevant foundation for developing an MPDSR implementation framework because it enables understanding of different levels and different factors that influence implementation including the intervention outcome as well as the implementation process.[35] Since not all constructs are applicable to M/PDSR implementation and some elements missing, we further built on CFIR drawing on the other frameworks and our experience of M/PDSR to develop the theory-based conceptual framework.

Box 1 presents the conceptual framework for this scoping review. It includes four domains: intervention (M/PDSR), outer setting, inner setting and individual. The first domain is related to the characteristics of the intervention being implemented into a particular setting. The complexity of M/PDSR as a process intervention with multi-faceted components and steps meant that we did not think two separate domains for intervention and process were needed and thus were combined. As with most interventions, there will be some adaptability at each level of M/PDSR as it is implemented in different settings and at different levels. Factors within the intervention domain for M/PDSR may include the steps of the audit cycle, cost and funding for the process, perceived legitimacy of the process as resulting in change and the perceived ability to test, adapt and implement it. The next two domains, the inner and outer setting, continuously interface and influence each other; thus the line between them is not always clear. The outer setting includes factors external to the organisation that influence implementation of M/PDSR; whereas the inner setting includes factors internal to the organisation. As outer setting factors influence implementation, change occurs in the inner setting. For M/PDSR implementation, the outer setting factors include policy and planning, linkages to other actors (such as professional association), pressures to implement, community links and communication channels. For the inner setting, implementation factors include readiness to implement, the structural characteristics of the organisation implementing M/PDSR, the organisational culture, the quality of communication and relationships and engagement of agents of change (also called champions in some settings). The last domain considers the characteristics of the individuals involved in implementation. Factors include their individual capacity and knowledge, their motivations and commitments to the implement M/PDSR, as well as their commitment to

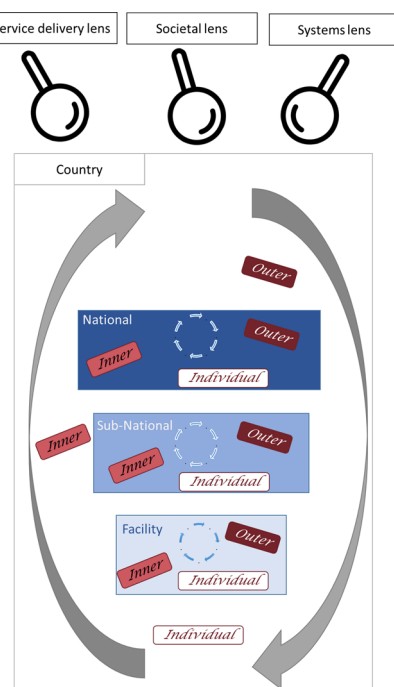

**Figure 1** Conceptual framework for implementation of M/PDSR. M/PDSR, maternal and/or perinatal death surveillance and response.

the team or organisation, and their willingness towards adapting to the intervention. The specific factors considered within each domain also vary depending on the level of implementation and on the context of the implementation effort.

The final component of the theoretical conceptual framework considers three different framings or lenses through which to understand and measure health system drivers of women's and children's health.[40] A service delivery lens includes the tangible inputs needed for M/PDSR implementation; a societal lens includes constructs that focus on social understanding and relationships; and a systems lens includes constructs that emphasise change dynamics which entails adaptive learning to contexts in ways that are not always anticipated. These three lenses have been presented by George *et al* as a way to describe both the tangible and intangible health systems drivers.[40] For each domain, we have categorised the constructs by these lenses.

We will test the framework on up to five different types of papers identified during the screening process and consider if any revisions need to be made. We will also undertake a consultation with experts in M/PDSR and implementation research to acquire their feedback and consider their recommendations for inclusion.

Figure 1 visualises application of the proposed conceptual framework to the concept of M/PDSR. As an intervention, M/PDSR is presented by the grey arrows encompassing various health systems levels and implementation factors that interact dynamically. Within a country, we acknowledge the multiple health system levels—national, sub-national, facility—through which

M/PDSR is operating. Within each level, there is a process for assessing the information relating to maternal and perinatal deaths (shown by the continuous circle). At each health systems level, there are different types of factors influencing implementation—outer, inner or individual. Finally, there are multiple lenses from which to understand and assess implementation (service delivery, societal and systems). For the scoping review, we will extract data with consideration of these multiple levels and factors.

### Scoping review protocol design

A scoping review was selected given the need for flexibility to explore different types of studies; and because it will facilitate a mapping and synthesis of available literature assessing implementation of M/PDSR and factors influencing implementation.[41]

The design will be guided by methods developed by Arksey and O'Malley[42] and expanded by Lavac et al[43] with guidance from the Joanna Briggs Institute (JBI) on conducting scoping reviews.[44] Details for the proposed six stages of a scoping review are described.

### Stage 1: identifying the research question

Our main research question is: 'What do we know about implementation of maternal death audit, perinatal death audit or combined audit approaches and the factors that either support or hinder the implementation process?' We also seek to answer: 'How can a theory-based conceptual implementation framework help to explain the various influencing implementation factors and their interactions?'

### Stage 2: identify relevant studies
*Process of search strategy*

The study will consider all literature that reports on implementation of maternal and/or perinatal death audit published in English between 2004 and July 2018 from LMIC. The start year is selected to coincide with the first WHO maternal death review guideline.[15]

We will include both quantitative and qualitative research studies. Peer-review publications will be primary sources but other published and unpublished (grey) literature such as reviews, reports and commentaries will also be taken into consideration.

An initial limited search of three online databases, which are relevant to our topic, will be undertaken using Google Scholar and PubMed to pilot the search strategy terms. Medical subject heading terms from PubMed will be used at the start to determine the words used to search in PubMed. We will combine search terms focussed on maternal mortality, perinatal mortality, audit/review systems and attributes of audit/review systems (search strategy found in online supplementary file 2).

After the initial search, we will analyse the text words contained in the title and abstract of retrieved articles, and of the index terms used to describe the articles. Revisions of our search strategy will be considered based on

**Table 1** Inclusion criteria for the scoping review

| Components | Application to this scoping review |
|---|---|
| Concept component | ► All forms of maternal and perinatal death review including obstetric audit, MPDSR, MDSR, MDR.<br>► Limited to studies or perspectives that identify factors that influence the implementation process. |
| Context component | ► Limited to low-income and middle-income countries listed by the World Bank in 2018. |

MDR, maternal death review; MDSR, maternal death surveillance and response; MPDSR, maternal and perinatal death surveillance and response.

the findings of the initial search and incorporate additional keywords, sources and search terms as appropriate. A second search using all the identified keywords and the index terms specific to each database will be undertaken across accessible databases and websites. The search will then be performed using the following additional electronic databases and online search engines: PubMed, CINAHL, SCOPUS, Web of Science, JSTOR, LILACS, the WHO Library, Maternal Death Surveillance and Response Action Network and Google.

The reference lists of all identified reports and articles will be searched for additional studies. All identified studies will be added into EndNote software and duplicate citations will be removed. We may contact the authors of primary studies or reviews for further information if necessary to provide clarity or to access additional information.

Finally, we will consult with experts in the field, including members of the WHO's MPDSR Technical Working Group, to ensure we have identified all relevant literature (published and grey).

*Characteristics of criteria*

Table 1 provides the inclusion criteria for this scoping review. For the concept component, we will only include literature that focusses on maternal or perinatal death reviews; thus excluding verbal autopsy or community death reviews, near-miss reviews, or confidential enquiries into maternal deaths. We also will exclude literature that does not specifically describe influencers of M/PDSR implementation. For example, some studies focus on the results of the audit data such as assessing cases of pre-eclampsia. If the article does not include factors exploring the implementation process, it will not be included.

### Stage 3: study selection
*Process of screening and data extraction*

Two reviewers will independently screen titles and then abstracts to check for relevance to the review. The reviewers will regularly meet during the screening process to discuss their selection of articles and to refine screening, if needed. In the cases where abstracts are not

available, the full text will be screened. All discrepancies between reviewers will be resolved by a third party.

Using the same process, the same reviewers will subsequently screen remaining full texts. All discrepancies between reviewers will be resolved by an independent third party.

### Stage 4: data collection

A data collection instrument will be developed by the research team according to the JBI guidance. The extracted data will include study characteristics (eg, type of reference such as article, report, study population, setting, study time period, study objective, study design). We will also consider the level or cultural factors addressed that is, policies, law and rhetoric (macro level), internalised routine practice at the sub-national and facility levels (meso level) and individual behaviour change (micro level).[6]

For describing implementation of M/PDSR, extracted data will be based on the constructs identified in the conceptual framework. The draft data extraction components are provided in online supplementary file 3. We will pilot test the data extraction tool during a workshop and agree on any revisions. The same team members who will undertake the screening process will extract data from the selected articles. The team will engage in weekly meetings to discuss any issues or questions relating to the extraction process; decisions on extraction process will be documented. The charting table may be updated if other additional unforeseen data are identified as extraction moves forward.

### Stage 5: data summary and synthesis of results

The review decision process will be reported using an adapted 'Preferred Reporting Items for Systematic Reviews and Meta-Analyses' extension for scoping reviews flow diagram.[45] Data analysis will involve qualitative content and thematic analysis linked to the conceptual framework.[46]

### Stage 6: consultation

Stakeholders will be engaged throughout the scoping review from helping to identify literature, to providing input on the conceptual framework and reviewing the findings to support interpretation. Consultations will be targeted at experts serving on the WHO's MPDSR Technical Working Group, as well as the Countdown to 2030 Drivers Technical Working Group. Other experts will be identified through a snowballing approach.

### Patient and public involvement

Given this is a protocol for a scoping review, patients and public were not involved in the design or research of the study.

### Proposed timeline

The process for conceptualising the scoping review, including the framework, began in April 2018. From September 2018 to March 2019, we began the consultation process with key stakeholders as well as the screening process. Data collection began in April 2019. We expect the scoping review will be completed in the first quarter of 2020.

## ETHICS AND DISSEMINATION

This study will contribute to the work of the Countdown to 2030 Drivers Group, which explores key contextual drivers for equitable and effective coverage of maternal and child health interventions as well as the MPDSR Technical Working Group co-led by the United Nations Population Fund (UNFPA), UNICEF and the WHO, which is tasked with advancing implementation of the intervention. This scoping review seeks to contribute specifically to the understanding how implementation of M/PDSR can drive quality improvements in service delivery responses to women and children's health. Limitations of the study include the parameters of the search criteria, notably around language (English only) and time span (from 2004 to July 2018) and search process (eg, not all grey literature will be identified). Because of these limitations, some literature or components of the identified literature may not be included in the results of this scoping review. Ethics approval is not required since the scoping review methodology consists of reviewing and collecting data from publicly available sources. We plan to publish the results of the scoping review in an academic journal as well as present to key stakeholders through various forum (ie, webinars, conferences, meetings). Consultation with key stakeholder groups will further guide dissemination efforts.

**Contributors** MVK conceived of the idea, developed the research question and study methods and contributed meaningfully to the drafting and editing; she also approved the final manuscript. ASG, DRW, PW and PW aided in developing the research question and study methods, contributed meaningfully to the drafting and editing, and approved the final manuscript.

**Funding** ASG and MVK are supported by the South African Research Chair's Initiative of the Department of Science and Technology and National Research Foundation of South Africa (Grant No 82769), the South African Medical Research Council and the Countdown 2030 project funded by the Bill and Melinda Gates Foundation. Any opinion, finding and conclusion or recommendation expressed in this material is that of the author and funders do not accept any liability in this regard.

**Competing interests** None declared.

**Patient consent for publication** Not required.

**Provenance and peer review** Not commissioned; externally peer reviewed.

**ORCID iD**
Mary V Kinney http://orcid.org/0000-0002-2903-0161

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
