## [Reviewer comments · BMJ Open]

ARTICLE DETAILS

TITLE (PROVISIONAL)	Implementation of maternal and perinatal death reviews: a scoping review protocol
AUTHORS	Kinney, Mary; Walugembe, David; Wanduru, Phillip; Waiswa, Peter; George, Asha S.

VERSION 1 – REVIEW

REVIEWER	Gabriel Dimitriou, MD, PhD Professor in Paediatrics & Neonatology University of Patras Head of the Paediatric Department-NICU-PICU Neonatal Intensive Care Unit, 3rd floor, University General Hospital of Patras RIO 26504 PATRAS GREECE
REVIEW RETURNED	01-Jun-2019

GENERAL COMMENTS	This paper presents a theory-based conceptual framework for understanding MPDSR implementation as well as a scoping review protocol to understand factors influencing MPDSR implementation in low-and middle-income countries. It is well written, yet rather difficult for the average reader to follow. The methodology is well described and referenced. The conceptual framework is well described and analysed. The implementation of this work will definitely offer a better insight in the field.
---

REVIEWER	Prof JJHM ERwich Department of Obstetrics and Gynecology University Medical Center Groningen University of Groningen Homepostcode CB20 PO Box 30 001 9700 RB Groningen The Netherlands Phone +31 50 3613008 tracer 43016 E-mail: j.j.h.m.erwich@umcg.nl
REVIEW RETURNED	20-Jun-2019

GENERAL COMMENTS	This paper describes a protocol for further study on the implementation of maternal and perinatal death reviews in LMIC. This is a much needed topic to study, the protocol contains several conceptions which makes it worthwhile publishing on its own. I am looking forward to the results.
--

REVIEWER	Christina Tiyankhulenji Mathias University of KwaZulu-Natal School of Nursing and Public Health Medicine Dept of Public Health and Medicine Durban, South Africa
REVIEW RETURNED	25-Jul-2019

GENERAL COMMENTS	INTRODUCTION -I suggest the phenomenon (M/PDSR) of the study should be captured/narrated in the first paragraph. As it is, the first two paragraphs are emphasizing on improving quality of care instead of emphasizing on M/PDSR which is one of the measures to improve quality care, which the phenomenon of interest in this review. -I suggest the introduction should show the impact of M/PDRS on perinatal and maternal mortality. This will define the importance of the phenomenon of the study on perinatal and maternal care, which will facilitate the linkage between the study results and the emphasize of the implementation of the M/PDRS as one of the measures to improve quality care. METHODOLOGY AND ANALYSIS -Line 35-54 there is no citation, please cite. STRENGTH AND LIMITATIONS OF THE STUDY -Consider forecasting limitations of the study
--

REVIEWER	MATSUI, Mitsuaki Nagasaki University School of Tropical Medicine and Global Health
REVIEW RETURNED	06-Aug-2019

GENERAL COMMENTS	I would request the authors to add one point. Authors mentioned that "we determined the Consolidated Framework for Implementation Research (CFIR) was the most relevant foundation for developing a MPDSR implementation framework because it enables understanding of different levels and different factors that influence implementation including the intervention outcome as well as the implementation process." It would be better to put additional explanation on CFIR in order to persuade the readers.
---

REVIEWER	Robert West University of Leeds, UK
REVIEW RETURNED	24-Sep-2019

GENERAL COMMENTS	This manuscript is very clearly written. The authors are following the recommendations of the Joanna Briggs Institute for their scoping review and as a consequence I see no issues arising. AS a minor point, the authors might consider the consistent use of -ize spelling. There are a couple of instances of -ise: internalise and conceptualise, organisations.
---

VERSION 1 – AUTHOR RESPONSE

Peer Review comments and responses for “Implementation of maternal and perinatal death reviews: a scoping review protocol”

COMMENTS	RESPONSES
Reviewer: 1 Reviewer Name: Gabriel Dimitriou, MD, PhD Institution and Country: Professor in Paediatrics & Neonatology University of Patras Head of the Paediatric Department-NICU-PICU Neonatal Intensive Care Unit, 3rd floor, University General Hospital of Patras RIO 26504 PATRAS GREECE	
This paper presents a theory-based conceptual framework for understanding MPDSR implementation as well as a scoping review protocol to understand factors influencing MPDSR implementation in low-and middle-income countries. It is well written, yet rather difficult for the average reader to follow. The methodology is well described and referenced. The conceptual framework is well described and analysed. The implementation of this work will definitely offer a better insight in the field.	Thank you for reviewing the paper and providing this feedback. We have reviewed the text to ensure clarity of concepts, which hopefully make it more accessible.
Reviewer: 2 Reviewer Name: prof JJHM ERwich Institution and Country: Department of Obstetrics and Gynecology University Medical Center Groningen University of Groningen Homepostcode CB20 PO Box 30 001 9700 RB Groningen The Netherlands Phone +31 50 3613008 tracer 43016 E-mail: j.j.h.m.erwich@umcg.nl	
This paper describes a protocol for further study on the implementation of maternal and perinatal death reviews in LMIC. This is a much needed topic to study, the protocol contains several conceptions which makes it worthwhile publishing on its own. I am looking forward to the results.	Thank you for reviewing the paper and providing this feedback.
Reviewer: 3 Reviewer Name: Christina Tiyankhuleni Mathias Institution and Country: University of KwaZulu-Natal School of Nursing and Public Health Medicine Depart of Public Health and Medicine Durban, South Africa	

INTRODUCTION -I suggest the phenomenon (M/PDSR) of the study should be captured/narrated in the first paragraph. As it is, the first two paragraphs are emphasizing on improving quality of care instead of emphasizing on M/PDSR which is one of the measures to improve quality care, which the phenomenon of interest in this review.	Thank you for reviewing the paper and providing this feedback. We have readjusted the background section to introduce M/PDSR first as proposed.
-I suggest the introduction should show the impact of M/PDRS on perinatal and maternal mortality. This will define the importance of the phenomenon of the study on perinatal and maternal care, which will facilitate the linkage between the study results and the emphasize of the implementation of the M/PDRS as one of the measures to improve quality care.	Thank you for this feedback. We added a sentence about the impact with references to the related Cochrane review and other impact studies.
METHODOLOGY AND ANALYSIS -Line 35-54 there is no citation, please cite.	Thank you for noting this gap. We have added more references to the section "Theory-based Implementation framework" as proposed.
STRENGTH AND LIMITATIONS OF THE STUDY -Consider forecasting limitations of the study	We have added this to the paper.
Reviewer: 4 Reviewer Name: MATSUI, Mitsuaki Institution and Country: Nagasaki University School of Tropical Medicine and Global Health	
I would request the authors to add one point. Authors mentioned that "we determined the Consolidated Framework for Implementation Research (CFIR) was the most relevant foundation for developing a MPDSR implementation framework because it enables understanding of different levels and different factors that influence implementation including the intervention outcome as well as the implementation process." It would be better to put additional explanation on CFIR in order to persuade the readers.	Thank you for reviewing the paper and providing this feedback. We have added more text explaining why CFIR was selected among the implementation frameworks assessed.
Reviewer: 5 Reviewer Name: Robert West Institution and Country: University of Leeds, UK	
This manuscript is very clearly written. The authors are following the recommendations of the Joanna Briggs Institute for their scoping review and as a consequence I see no issues arising. AS a minor point, the authors might consider the consistent use of -ize spelling. There are a couple of instances of -ise: internalise and conceptualise, organisations.	Thank you for reviewing the paper and providing this feedback. We have corrected the spelling.

VERSION 2 – REVIEW

REVIEWER	Mitsuaki MATSUI Nagasaki University School of Tropical Medicine and Global Health, Nagasaki, JAPAN
REVIEW RETURNED	27-Oct-2019
GENERAL COMMENTS	The revised manuscript is well written with clear explanation from the authors.